# Trends in Overall Survival and Treatment Patterns in Two Large Population-Based Cohorts of Patients with Breast and Colorectal Cancer

**DOI:** 10.3390/cancers11091239

**Published:** 2019-08-23

**Authors:** Doris van Abbema, Pauline Vissers, Judith de Vos-Geelen, Valery Lemmens, Maryska Janssen-Heijnen, Vivianne Tjan-Heijnen

**Affiliations:** 1Department of Internal Medicine, GROW-School for Oncology and Developmental Biology, Maastricht University Medical Center, Peter Debyelaan 25, 6229 HX Maastricht, The Netherlands; 2ACHIEVE Centre of Applied Research, Faculty of Health, Amsterdam University of Applied Sciences, Tafelbergweg 51, 1105 BD Amsterdam, The Netherlands; 3Department of Research, Netherlands Comprehensive Cancer Organisation (IKNL), Godebaldkwartier 419, 3511 DT Utrecht, The Netherlands; 4Department of Public Health, Erasmus Medical Center, Wytemaweg 80, 3015 CN Rotterdam, The Netherlands; 5Department of Clinical Epidemiology, VieCuri Medical Centre, Tegelseweg 210, 5912 BL Venlo, The Netherlands; 6Department of Epidemiology, GROW-School for Oncology and Developmental Biology, Maastricht University Medical Centre, Universiteitssingel 60, 6229 ER Maastricht, The Netherlands

**Keywords:** breast cancer, colorectal cancer, relative survival, older patients, geriatric oncology, cancer treatment

## Abstract

Previous studies showed substantial improvement of survival rates in patients with cancer in the last two decades. However, lower survival rates have been reported for older patients compared to younger patients. In this population-based study, we analyzed treatment patterns and the survival of patients with breast cancer (BC) and colorectal cancer (CRC). Patients with stages I–III BC and CRC and diagnosed between 2003 and 2012 were selected from the Netherlands Cancer Registry (NCR). Trends in treatment modalities were evaluated with the Cochran-Armitage trend test. Trends in five-year overall survival were calculated with the Cox hazard regression model. The Ederer II method was used to calculate the five-year relative survival. The relative excess risk of death (RER) was estimated using a multivariate generalized linear model. During the study period, 98% of BC patients aged <75 years underwent surgery, whereas for patients ≥75 years, rates were 79.3% in 2003 and 66.7% in 2012 (*p* < 0.001). Most CRC patients underwent surgery irrespective of age or time period, although patients with rectal cancer aged ≥75 years received less surgery or radiotherapy over the entire study period than younger patients. The administration of adjuvant chemotherapy increased over time for CRC and BC patients, except for BC patients aged ≥75 years. The five-year relative survival improved only in younger BC patients (adjusted RER 0.95–0.96 per year), and was lower for older BC patients (adjusted RER 1.00, 95% Confidence Interval (CI) 0.98–1.02, and RER 1.00; 95% CI 0.98–1.01 per year for 65–74 years and ≥75 years, respectively). For CRC patients, the five-year relative survival improved over time for all ages (adjusted RER on average was 0.95 per year). In conclusion, the observed survival trends in BC and CRC patients suggest advances in cancer treatment, but with striking differences in survival between older and younger patients, particularly for BC patients.

## 1. Introduction

Together with the aging of the population in Western countries, the number of older people with cancer has increased considerably in the last decade [1]. In the Netherlands, breast cancer (BC) is the most common cancer in women, with an estimated incidence rate of 66 new cases per 100,000 women [2]. The incidence of BC is highest in patients aged 70–74 years (212 per 100,000 women) [2]. One of the most common cancers in both men and women is colorectal cancer (CRC), with an estimated incidence of 65 new cases per 100,000 persons. Its incidence increases with age (415 per 100,000 per persons in those aged 75–79 years) [2].

Though the group of older patients has increased considerably, evidence to guide treatment of these patients remains limited [3]. Many clinical trials exclude older patients from participating [4], and older patients who do participate in clinical trials may not be representative of the general older population, as clinical trials often exclude older patients with comorbidities or poor overall health [5].

Several population-based studies showed that the survival rate of BC patients has improved substantially in the last two decades [1,6,7,8]. This has been attributed particularly to mammographic screening programs [9] and advances in treatment (e.g., improved radiotherapy techniques and new systemic therapeutic agents such as third-generation chemotherapy, aromatase inhibitors, and HER2-targeted therapies). Clinical guidelines, as well as international position papers, state that surgery is recommended for early-stage BC patients [3,10]. Adjuvant chemotherapy is recommended for node-positive or high-risk node-negative disease, and endocrine therapy is recommended for high-risk receptor-positive disease [3,10]. However, population-based studies showed that older BC patients are less likely to receive standard care, which has been linked to lower survival rates in older patients [6,8,11].

Surgery is recommended in early-stage CRC patients. To reduce the risk of recurrence after surgery, clinical guidelines recommend that all lymph node-positive patients should be considered for adjuvant treatment, e.g., radiotherapy, or chemotherapy [12,13]. Several population-based studies showed that the administration of adjuvant chemotherapy has increased considerably in recent years, and that this is associated with improved survival [14,15]. However, the administration of adjuvant chemotherapy is considerably less common in older patients [14].

Several studies state that age should not be a contraindication for surgery or adjuvant chemotherapy [16]. Trends in survival and treatment have been published before, but this has not clarified the differences in survival rates between cancers. The aim of this study was to investigate trends in survival rates and treatment patterns over time for older versus younger patients with BC and CRC in the Netherlands.

## 2. Results

Between 2003 and 2012, 127,146 patients were diagnosed with stages I–III BC in the Netherlands, while 85,629 patients were diagnosed with stages I–III CRC (Table 1). The median age of patients with BC was 60 years (range 18 to 103), and the median age of patients with CRC was 71 years (range 18 to 102).

### 2.1. Stage at Diagnosis

The stages at diagnosis according to age are presented in Figure 1. BC patients aged ≤45 or ≥75 years were more often diagnosed with stage II or III cancer compared to those aged 45–75 years (*p* < 0.001). Stage II was diagnosed in 51.7% and stage III in 17.9% of the BC patients aged ≥75 years. The percentage of CRC patients with stage II disease steadily increased with age (32.5% in CRC patients aged 45–54 years, and 44.3% in CRC patients aged ≥75 years), while stage III decreased with age (46.7% in CRC patients aged 45–54 years, and 33.3% in CRC patients aged ≥75 years, *p* < 0.001).

### 2.2. Trends in Treatment

The time trends for all treatments for BC patients between 2003 and 2012 are presented in Figure 2. Over the entire study period, 98.0% of patients with stages I–III BC who were younger than 75 years underwent surgery, whereas for those aged 75 years or over, surgery rates declined from 79.3% in 2003 to 66.7% in 2012 (*p* < 0.001, Figure 2a). The proportion of BC patients undergoing lymph node dissection for node-positive disease decreased over time for all age groups (Figure 2b). The use of radiotherapy after breast-conserving surgery increased significantly for all age groups but was lowest among patients aged 75 years and older (73.6% in 2003 vs. 84.1% in 2012, *p* < 0.001) and highest among patients aged 45–54 years (84.4% in 2003 vs. 96.2% in 2012, *p* < 0.001, Figure 2c). Primary endocrine therapy as a monotherapy in BC patients was almost exclusively used for patients aged 75 years or over, and increased significantly over time (16.5% in 2003 and 28.7% in 2012, *p* < 0.001, Figure 2d). The use of adjuvant chemotherapy in stages II and I BC patients increased significantly over time for all age groups, except for patients aged 75 years or older. Its use was associated with age, with more than 90% of patients younger than 55 years and less than 9% of patients aged 65 years or older in 2012 receiving it, and hardly any patients aged 75 years or older (Figure 2e,f).

As CRC patients are treated according to tumor location, i.e., colon cancer (CC) or rectal cancer (RC), we report the treatments received for CC and RC separately (Figure 3). In CC, the number of patients aged 75 years or older who underwent surgery declined significantly over time (98.1% in 2003 vs. 94.2% in 2012, *p* < 0.001, Figure 3a). The percentage of chemotherapy treatment in stage III CC patients aged 55 years or older increased significantly over time (Figure 3b), although the percentage of those aged 75 years or older who received chemotherapy remained considerably lower (16.3% in 2003 vs. 23.5% in 2012, *p* < 0.001). The percentage of RC patients aged 45 years or older who underwent surgery declined significantly over time, the greatest decrease being observed in RC patients aged 75 years or older (from 91.8% in 2003 to 81.1% in 2012, *p* < 0.001) (Figure 3c). The use of radiotherapy in stage III RC increased significantly over time for all age groups but became less frequent with increasing age (91.8% of patients aged 55–64 years compared to 80.2% of patients aged 75 years or older in 2012, Figure 3d). The number of RC patients aged 75 years or older undergoing surgery was lower than that of younger patients. The percentage of I-III RC patients aged 75 years or older who received primary radiotherapy as a single treatment modality increased significantly, from 3.8% in 2003 to 9.4% in 2013 (*p* < 0.001, Figure 3e). The use of chemoradiation in stage II RC patients increased significantly in all age groups, ranging from 19.8% to 71.6% in 2012, but again this was highly dependent on age, with the lowest use in the age group of 75 years and older (Figure 3f).

### 2.3. Trends in Survival

The 5-year overall survival of BC patients is presented in Figure 4. The 5-year overall survival improved in BC patients (adjusted hazard ratio (HR) 0.78, 95% CI 0.73–0.83) in the period 2003–2012. The 5-year overall survival improved slightly in patients aged 75 years or older (adjusted HR 0.99, 95% CI 0.98–0.99) and did not significantly improve in patients aged 65–74 years (adjusted HR 0.99, 95% CI 0.98–1.01). In the multivariable Cox regression analysis, the overall survival improved in the period 2003–2012 and was influenced significantly by age (Table 2). The 5-year relative survival improved in BC patients, from 89.7% in 2003 to 93.1% in 2012 (adjusted relative excess risk (RER) 0.97, 95% CI 0.97–0.98 per year) (Figure 5). The 5-year relative survival rate improved significantly for BC patients aged 65 years or younger over the period between 2003 and 2012. In contrast, the relative survival for BC patients aged 75 years or older did not increase significantly over the same period (crude RER 0.99, 95% CI 0.97–1.02). After adjustment for age, stage, and treatment, the RER for BC patients aged 65–74 years was no longer significant (adjusted RER 1.00, 95% CI 0.98–1.01), and the RER for patients aged 75 years or older remained non-significant (adjusted RER 1.00, 95% CI 0.98–1.01).

Similarly, the 5-year overall survival improved for all CRC patients (Figure 6). Overall survival in CRC patients was influenced by the year of diagnoses and age. Other factors affecting 5-year overall survival were, stage, grade, gender, and treatment. The relative survival for CRC patients improved from 76.0% in 2003 to 80.3% in 2012 (adjusted RER 0.95, 95% CI 0.94–0.95 per year, Figure 7). Most overall improvement was observed in the CRC patients aged 65–74 years. In this group, the 5-year relative survival increased by 7.9% between 2003 and 2012 (adjusted RER 0.94, 95% CI 0.93–0.95). In contrast to the BC cohort, the RERs for patients with CRC in all age groups had improved significantly after adjustment for grade, sex, stage, and treatment (adjusted RERs varying between 0.94 and 0.96 per year).

## 3. Discussion

The findings of our population-based study show that 5-year overall and relative survival improved for BC patients and for CRC patients in the Netherlands between 2003 and 2012. The difference in 5-year survival between older and younger CRC patients decreased during the study period, and the greatest improvement in overall and relative survival was observed for older CRC patients. Among BC patients, substantial improvement of 5-year relative survival was seen only for younger BC patients. The overall and relative survival and its improvement over time were lower for older BC patients.

Previous population-based studies found that overall and relative survival among older BC patients did not change in the last decade [17,18]. This lack of improvement has been attributed to differences in the delivery of cancer care between older and younger BC patients, e.g., the influence of comorbid conditions on the selection of cancer treatments [19,20]; the lower likelihood of receiving breast surgery, radiotherapy, or adjuvant chemotherapy for older BC patients compared to younger BC patients [6]; and mammographic screening being offered only to younger women [9]. The survival and treatment differences observed in our study suggest possible limitations of cancer treatment, as well as reticence about cancer treatment, for older BC patients.

The most notable finding was the decreasing proportion of older BC patients receiving surgery, and the increasing proportion receiving primary endocrine therapy. Previous population-based studies in the Netherlands confirm this finding [6,17,21]. Van de Glas et al. showed that the proportion of older BC patients that did not receive breast surgery increased substantially from 9.2% in 1995 to 30.1% in 2011 [17]. Moreover, previous population-based studies showed that the proportion of older BC patients receiving primary endocrine therapy is higher in the Netherlands than in other European countries [22,23,24]. The EUROCARE breast cancer group found lower survival rates in European countries, like England and Ireland, where older BC patients are treated more often with primary endocrine therapy omitted [22]. This increased use of primary endocrine therapy may have been influenced by a meta-analysis, in which seven randomized controlled trials were included [25]. This meta-analysis showed that, compared to those treated with primary endocrine therapy, surgery alone seemed to have no beneficial impact on the overall survival rate (HR 0.98, 95% CI 0.74–1.30). However, BC patients who underwent surgery with adjuvant endocrine therapy had a borderline significantly improved overall survival rate (HR 0.86, 95% CI 0.73–1.00) and a significantly better disease-free survival. The HR for endocrine therapy alone was 0.65 (95% CI 0.53–0.81) [25].

Recent reports have advocated that primary endocrine therapy should only be considered in frail older BC patients [26]. Guidelines in the Netherlands state that patients with operable tumors should be treated with surgery and not primary endocrine therapy, irrespective of age [10]; also, the International Society of Geriatric Oncology (SIOG) recommends that primary endocrine therapy should only be offered to BC patients with a life expectancy of less than three years, or who are unfit for or refuse surgery [3]. However, as a large proportion of older BC patients are currently being treated with primary endocrine therapy, it is not clear if this recommendation is being followed consistently. In a small retrospective study in the Netherlands, Hamakers et al. found that the reason for not carrying out surgery was not always clearly stated in medical charts. When documented, reasons were often higher age, poor overall health, patient preference, and comorbidities [21]. Additionally, Sierink et al. found in a small cohort in the Netherlands that BC patients are less likely to receive surgery when they are older, have been diagnosed with more comorbidities, or have been diagnosed with a more advanced disease [27]. However, other studies found that age was the most important reason for foregoing surgery in older BC patients, even when adjustments were made for comorbidities [28,29,30].

Several studies found that the main factor for surgical morbidity in BC patients are comorbidities and not age [31]. Breast surgery can be considered a low morbidity surgery [32]. Approximately one third of BC patients develop breast and/or axillar wound infections, seromas, and hematomas after surgery, depending on the type of surgery. These surgical morbidities are often minor and treated in an outpatient setting [32]. Moreover, van de Glas et al. demonstrated that relative mortality was not influenced by postoperative complications in older BC patients [33].

Our study showed that the proportion of surgery in older CRC patients remains high. Many studies have shown that age should not prohibit surgery in CRC patients [16]. In RC, however, the proportion of patients aged ≥75 years that received primary chemoradiation increased from 0% in 2003 to almost 20% in 2012. Recently, foregoing surgery after chemoradiation has been shown to be safe for RC patients in whom a complete clinical response is seen [34].

In the present study, the proportion of older BC patients receiving adjuvant chemotherapy was lower than that of younger BC patients. In Europe, different rates of adjuvant chemotherapy in BC patients have been reported, with lower rates in the Netherlands [22]. There has been conflicting evidence regarding the benefits of this treatment for older BC patients. Data from the Early Breast Cancer Trial demonstrated a decreasing benefit from adjuvant chemotherapy with age [35,36], although relatively few BC patients aged 70 years or older have been included in clinical trials [5]. Nevertheless, the results of this meta-analysis may have contributed to the lower use of adjuvant chemotherapy in older BC patients [35]. Other recent studies suggest a possible benefit from adjuvant chemotherapy in older BC patients, especially in those with HER2-positive disease [37,38]. Recently, Du et al. compared the effectiveness of adjuvant chemotherapy between older BC and CRC patients, and found that chemotherapy might be effective in BC patients up to the age of 79 years [39].

Possible explanations why older BC patients receive adjuvant chemotherapy less often are frailty, poor functional status, or comorbidities. In addition, older patients are less willing to accept possible side effects associated with adjuvant chemotherapy compared with younger patients, and are concerned about negative influences of adjuvant chemotherapy. Most older BC patients receive adjuvant endocrine therapy because they are more likely to have a receptor positive disease, which tends to grow more slowly, are usually well differentiated, and respond better to endocrine therapy. However, there are side effects associated with endocrine therapy, like deep vein thrombosis or pulmonary emboli [40].

The 5-year overall and relative survival improved in older CRC patients in our study. In a population-based study, Van den Broek et al. showed that 5-year relative survival had increased significantly in CC patients aged <65 years and ≥75 years over the period from 1991 to 2005, but not in patients aged 65–74 years [41]. In our study the greatest overall improvement of relative survival was seen in CRC patients aged 65–74 years (74.4% in 2003 vs. 82.3% in 2012). In the ≥75 years age group, relative survival improved significantly, but still lagged behind the survival of younger age groups. However, the difference in survival between younger and older CRC patients reported by the EUROCARE working group [1,7] diminished over time in our study. This improved survival could be partly attributed to the increased use of adjuvant chemotherapy in older patients [14], but also to other factors, such as improved preoperative staging and improved surgery [42].

This study had some limitations. Information on patient characteristics, such as comorbidities, was not available. Furthermore, we can draw no conclusions about aspects such as quality of life, maintaining functional status, risk of recurrence, or complications, as we had no data for these outcomes. Although we adjusted the analyses for potential confounders, there may still have been some confounding by other factors. Finally, we did not have data on the causes of mortality. On the other hand, the use of data for two different tumor types means that our study does provide important insights into the differential treatment and survival trends by age. Another strength of our study is the large number of patients with BC and CRC included from the Netherlands Cancer Registry, with at least 5 years of follow-up.

## 4. Materials and Methods

### 4.1. Patients

For this population-based study, we selected all patients with primary BC and CRC stages I–II diagnosed between 2003 and 2012 from the Netherlands Cancer Registry (NCR). We only selected patients with stages I–III, as this group could potentially be treated with curative intent. Furthermore, only female BC patients were selected. The NCR records all newly diagnosed malignancies in the Netherlands after notification from the nationwide automated pathological archive (PALGA). This data is supplemented with data from the National Registry of Hospital Discharge Diagnosis. Trained registrars from the NCR routinely collect data from medical records regarding patient, tumor, and treatment characteristics in all Dutch hospitals.

Cancer stage was based on the pathological TNM classification applicable at the time of cancer diagnosis. In case of a missing pathological TNM classification, the clinical TNM classification was used instead. Data regarding the vital status of all patients was available through linkage with the municipal personal records database, with complete follow-up until 31 December 2017. Patients were divided into five groups according to age: 15–44, 45–54, 55–64, 65–74, and 75 years or older. Primary cancer treatment retrieved from the NCR included surgery, radiotherapy, endocrine therapy, immunotherapy, and (neo-)adjuvant chemotherapy.

According to the Central Committee on Research involving Human Subjects (CCMO), this type of study does not require approval from an ethics committee in the Netherlands. The Privacy Review Board of the Netherlands Cancer Registry approved this study in January 2018 (K17.351).

### 4.2. Statistical Analyses

All analyses were performed in IBM SPSS statistics version 22.0 (SPSS, Chicago, IL, USA) or SAS statistics version 9.4 (SAS institute, Inc., Cary, NC, USA). Statistical significance was defined as *p* ≤ 0.05.

The Cochran-Armitage trend test was used to test for differences in treatment over time. Treatments are presented as percentages per age group and period.

The Cox hazard analysis was used to calculate the overall survival. The hazard ratio (HR) was calculated according to age group and cancer stage, and adjusted for cancer stage, grade, gender, and treatment. 

As cause of death is not available in the NCR, relative survival was used as an estimation of the disease-specific survival, and was calculated using the Ederer II method, i.e., as the observed overall survival of patients in the study divided by the expected overall survival in a matched group of the general population by age, sex, and year [11]. The expected survival was obtained from national life tables. The relative excess risk of death (RER) was estimated using a multivariate generalized linear model with a Poisson distribution. The RER was calculated according to age and cancer type, and adjusted for cancer stage, grade, gender, and treatment. The treatment variables were added to investigate the effect of treatment on the RER.

## 5. Conclusions

In this population-based study analyzing oncological treatment patterns and relative survival of BC and CRC patients, we observed substantial differences between younger and older patients. Although the observed survival trends in BC and CRC patients suggest advances in cancer treatment, we found that the survival of older BC patients in particular was strikingly lower. Moreover, the differences in relative survival between younger and older BC patients revealed in previous studies were found to have continued to increase in recent years. Inequalities in the provision of cancer care to older BC patients need to be investigated in future cancer research. Selection criteria for specific treatments could eventually lead to individualized and optimized treatment for older cancer patients.

## Figures and Tables

**Figure 1 cancers-11-01239-f001:**
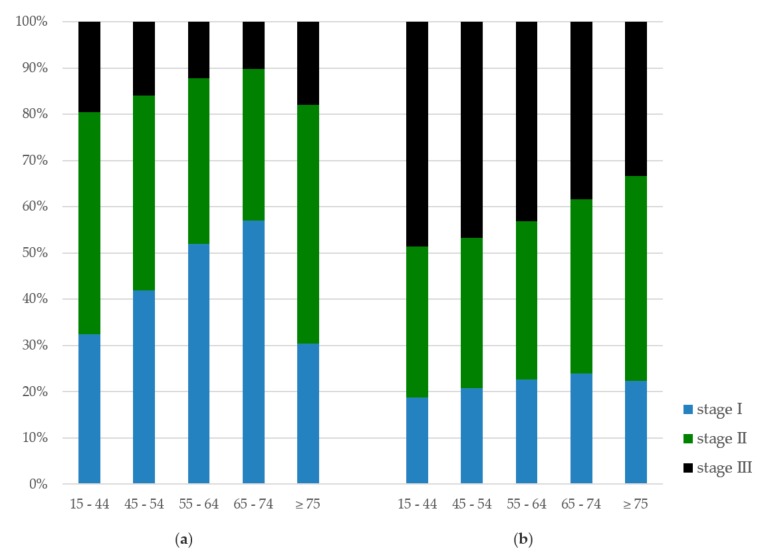
Tumor stage by age at diagnosis: (**a**) breast cancer, and (**b**) colorectal cancer.

**Figure 2 cancers-11-01239-f002:**
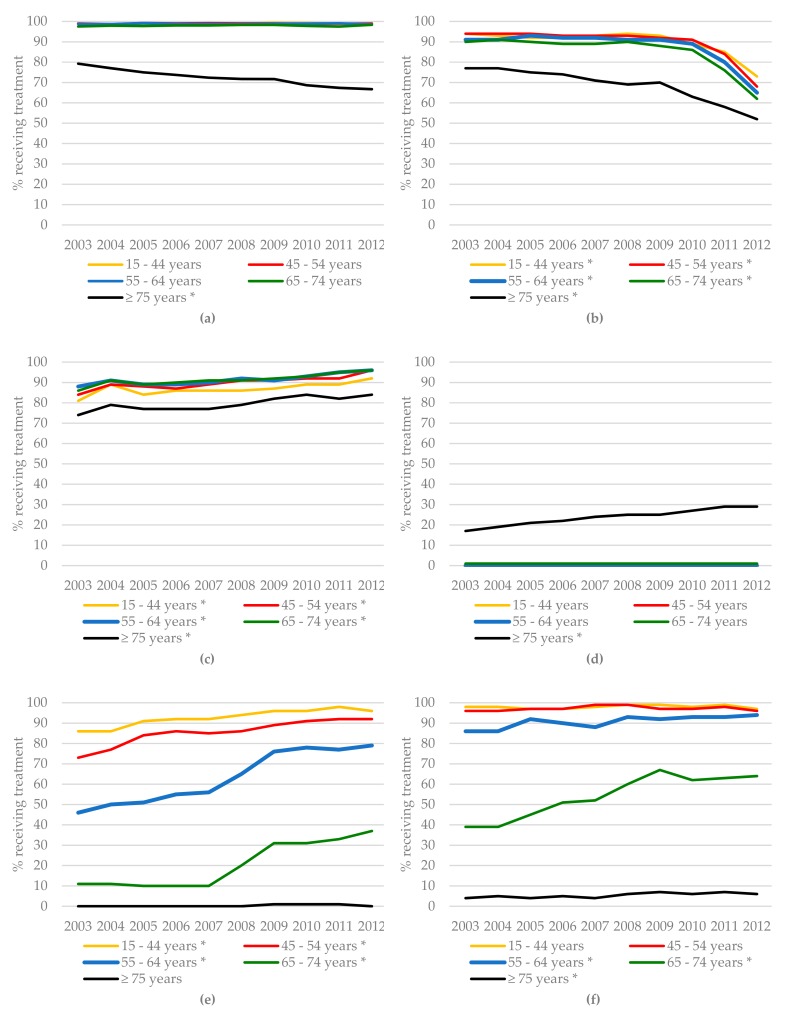
Time trends in the treatment of breast cancer (BC) patients according to age group: (**a**) surgery in BC patients stages I–III, (**b**) lymph node dissection in BC patients stages I–III with node-positive disease, (**c**) radiotherapy after breast-conserving surgery in BC patients stages I–III, (**d**) primary endocrine therapy in BC patients stages I–III, (**e**) chemotherapy in BC patients stage II, and (**f**) chemotherapy in BC patients stage III. * Significant (*p* < 0.05) difference in treatment over time using the Cochran–Armitage trend test.

**Figure 3 cancers-11-01239-f003:**
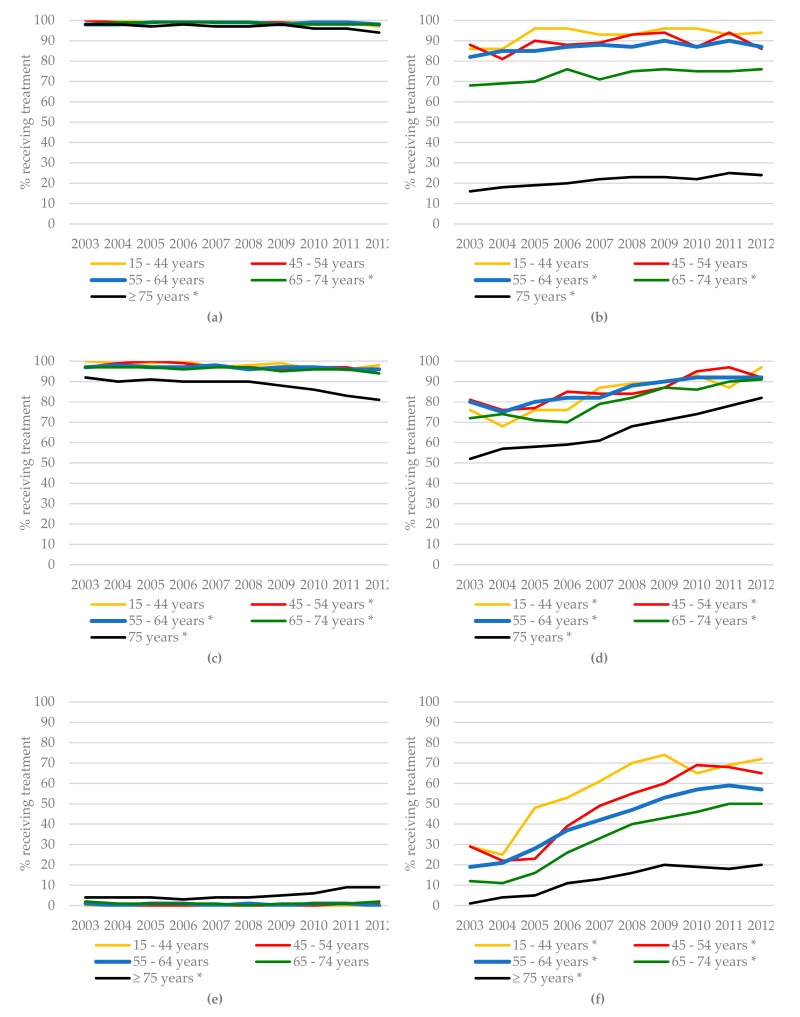
Time trends of treatment in colon cancer (CC) and rectal cancer (RC) patients according to age group: (**a**) surgery in CC patients stages I–III, (**b**) chemotherapy in CC patients stage III, (**c**) surgery in RC patients stages I–III, (**d**) radiotherapy in RC patients stage III, (**e**) primary radiotherapy as a single treatment modality in RC patients stage III, and (**f**) chemoradiation in RC patients stages II–III. * Significant (*p* < 0.05) difference in treatment over time using the Cochran–Armitage trend test.

**Figure 4 cancers-11-01239-f004:**
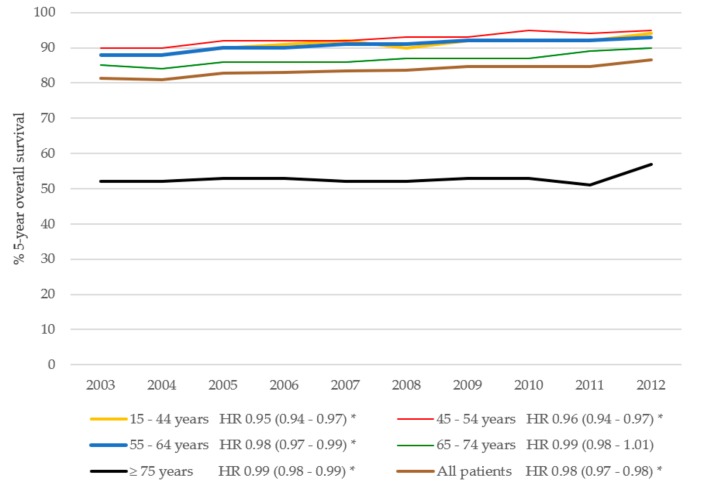
Five-year overall survival in BC patients over time per age group. HR indicates hazard ratio per year, adjusted for grade, stage, and treatment. * Significant (*p* < 0.05).

**Figure 5 cancers-11-01239-f005:**
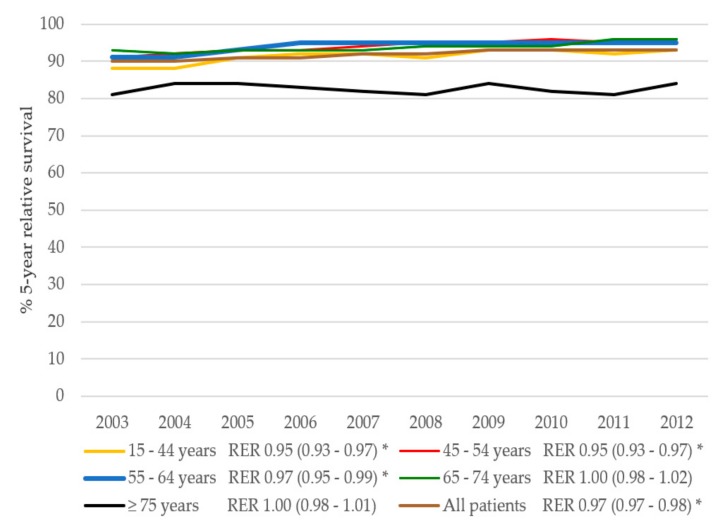
Five-year relative survival in BC patients over time per age group. RER indicates relative excess risk per year, adjusted for grade, stage, and treatment. * Significant (*p* < 0.05).

**Figure 6 cancers-11-01239-f006:**
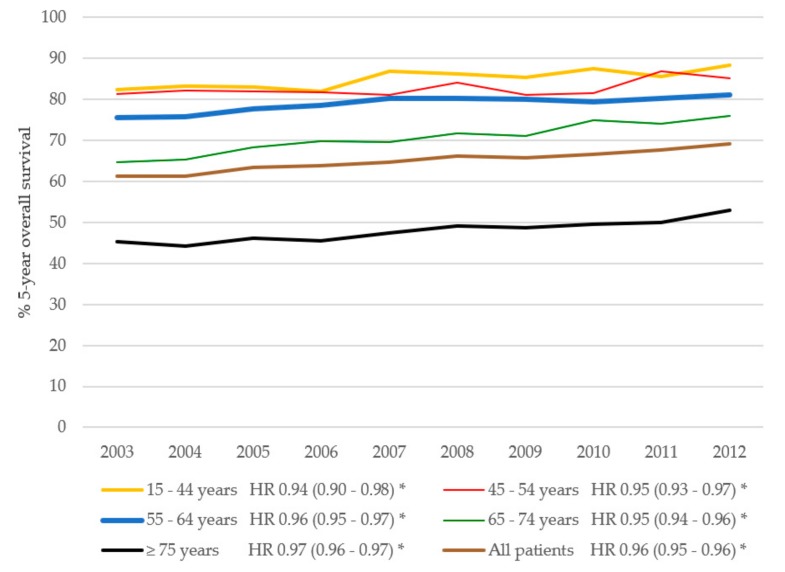
Five-year overall survival in CRC patients over time per age group. HR indicates the hazard ratio per year, adjusted for grade, stage, gender, and treatment. * Significant (*p* < 0.05).

**Figure 7 cancers-11-01239-f007:**
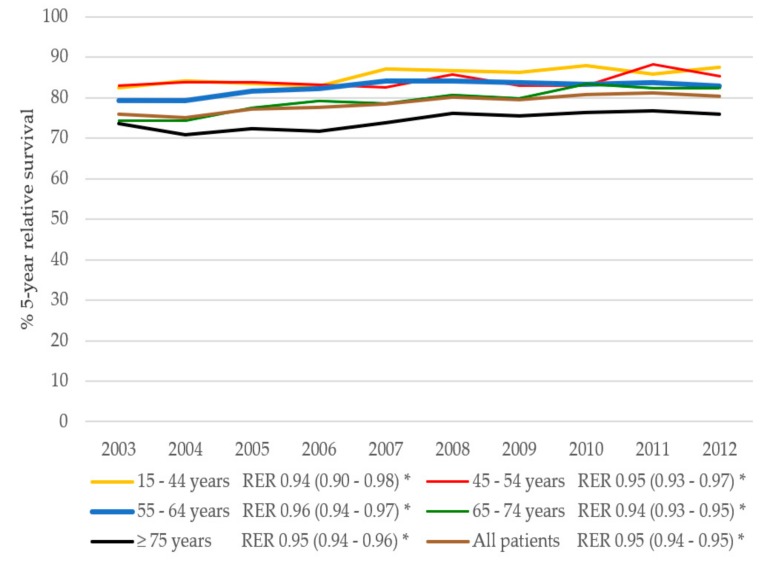
Five-year relative survival in CRC patients over time per age group. RER indicates the relative excess risk per year, adjusted for grade, stage, gender, and treatment. * Significant (*p* < 0.05).

**Table 1 cancers-11-01239-t001:** Clinical characteristics of 127,146 breast cancer (BC) and 85,629 colorectal cancer (CRC) patients, diagnosed in 2003–2012.

Characteristics	BC		CRC	
	*n*	%	*n*	%
Gender				
Male	-		45,783	53.5
Female	127,146	100	39,846	46.5
Age (years)				
15–44	15,206	12	2077	2.4
45–54	30,539	24	6457	7.5
55–64	31,830	25	17,969	21
65–74	27,095	21.3	26,839	31.3
≥75	22,476	17.7	32,287	37.7
Year of diagnosis				
2003	11,669	9.2	7367	8.6
2004	11,848	9.3	7813	9.1
2005	11,845	9.3	7877	9.2
2006	12,264	9.6	8190	9.6
2007	12,725	10	8510	9.9
2008	12,836	10.1	8713	10.2
2009	12,999	10.2	8877	10.4
2010	13,102	10.3	9135	10.7
2011	13,838	10.9	9486	11.1
2012	14,020	11	9661	11.3
Stage				
I	56,579	44.5	19,453	22.7
II	52,049	40.9	33,363	39
III	18,518	14.6	32,813	38.3

**Table 2 cancers-11-01239-t002:** Multivariate Cox hazard analysis of five-year overall survival.

Variable	BC		CRC	
	HR (95% CI) ^a^	*p*	HR (95% CI) ^a^	*p*
Year of diagnosis (per year)	0.98 (0.97–0.98)	<0.001	0.96 (0.96–0.97)	<0.001
Age		<0.001		<0.001
15–44 years	1.04 (0.97–1.11)		**0.87 (0.76–0.98)**	
45–54 years	1.00 (ref)		1.00 (ref)	
55–64 years	**1.35 (1.28–1.43)**		**1.25 (1.17–1.34)**	
65–74 years	**2.06 (1.95–2.18)**		**1.81 (1.70–1.93)**	
≥75 years	**4.89 (4.60–5.18)**		**3.43 (3.22–3.65)**	
Gender	-			<0.001
Male	-		1.00 (ref)	
Female			**0.82 (0.80–0.84)**	
Stage		<0.001		<0.001
I	1.00 (ref)		1.00 (ref)	
II	**1.79 (1.72–1.87)**		**1.50 (1.44–1.55)**	
III	**4.14 (3.96–4.32)**		**2.74 (2.64–2.84)**	
Grade		<0.001		<0.001
Well differentiated	1.00 (ref)		1.00 (ref)	
Moderately differentiated	**1.29 (1.22–1.35)**		1.02 (0.97–1.07)	
Poorly differentiated and	**2.14 (2.03–2.25)**		**1.46 (1.38–1.55)**	
undifferentiated				
Unknown	**1.67 (1.57–1.77)**		**1.12 (1.06–1.19)**	
Surgery		<0.001		<0.001
No	1.00 (ref)		1.00 (ref)	
Yes	**0.30 (0.28–0.32)**		**0.14 (0.14–0.15)**	
Radiotherapy		<0.001		<0.001
No	1.00 (ref)		1.00 (ref)	
Yes	**0.63 (0.61–0.65)**		**0.76 (0.74–0.78)**	
Chemotherapy		<0.05		<0.001
No	1.00 (ref)		1.00 (ref)	
Yes	**0.95 (0.90–0.99)**		**0.64 (0.62–0.66)**	
Hormone therapy		<0.001		
No	1.00 (ref)		-	
Yes	**0.55 (0.53–0.57)**			

^a^ Adjusted for all variables included in the model. HR, hazard ratio. Significant (*p* < 0.05) HR values are indicated in bold. ref, reference 95% CI, 95% Confidence Interval.

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
