# Peer review of "Trends in Overall Survival and Treatment Patterns in Two Large Population-Based Cohorts of Patients with Breast and Colorectal Cancer"

_cancers, 2019, doi:10.3390/cancers11091239_

Round 1

Reviewer 1 Report

I suggest to add some more comments that distinguish endocrine therapy from chemotherapy in patients with breast cancer. Accordingly, the following paper should be taken into account:

Crivellari D, et al. Crit Rev Oncol Hematol. 2010 Jan;73(1):92-8.

In order to analyze deeply the reasons that could have determined the different approaches in elderly and younger patients, Authors should spend some more comments to describe the surgical treatment for breast and colorectal cancer.

In addition, they should provide some information about the change of systemic (adjuvant) treatment overtime. In fact, it should be noted that patients who did not undergo surgery were also deprived of adjuvant therapy. Some comments about this thoughts should be added in the discussion

Reviewer 2 Report

The authors have presented a manuscript examining the differences in trends in overall survival and treatment patterns for breast and colorectal cancer. There are concerns for the authors' consideration in order to improve the quality of the paper.

1. Lines 22-23, and 70-72: The rationale for this project is not compelling. A stronger rationale should be proposed.

2. The introduction should be revised to emphasize why CRC survival is being compared with BC survival, since the authors have cited previously published works in the introduction.

3. Line 221:Since only female BC patients were included in the study, one would have expected that only female CRC patients will have been included too, in order to reduce any errors in result interpretation from confounders such as sex hormones. Survival can be greatly impacted by the sex of an individual. Table 1 shows that there were more males (53.5%) in the CRC group. How did you account for this in your study?

4.  The Ederer II Method and the Multivariate GLM were used to calculate 5 year RS and RER for individual age groups. An observed difference in survival of individual groups over time does not necessarily mean a true difference between group survival. A Cox regression model is required to determine if there are true differences in survival between the age groups. 

Author Response

Reviewer #2

The authors have presented a manuscript examining the differences in trends in overall survival and treatment patterns for breast and colorectal cancer. There are concerns for the authors' consideration in order to improve the quality of the paper.

1.       Lines 22-23, and 70-72: The rationale for this project is not compelling. A stronger rationale should be proposed.

We revised the introduction of the manuscript

Introduction, page 2

Together with the aging of the population in Western countries, the number of older people with cancer has increased considerably in the last decade [1]. In the Netherlands, breast cancer (BC) is the most common cancer in women, with an estimated incidence rate of 66 new cases per 100,000 women [2]. The incidence of BC is highest in patients aged 70–74 years (212 per 100,000 women) [2]. One of the most common cancers in both men and women is colorectal cancer (CRC), with an estimated incidence of 65 new cases per 100,000 persons. Its incidence increases with age (415 per 100,000 per persons in those aged 75–79 years) [2]. Though the group of older patients has increased considerable, evidence to guide treatment of these patients remains limited [3]. Many clinical trials exclude older patients from participating [4], and older patients who do participate in clinical trials may not be representative of the general older population, as clinical trials often exclude older patients with comorbidities or poor overall health [5].

Several population-based studies showed that the survival rate of BC patients has improved substantially in the last two decades [1,6-8]. This has been attributed particularly to mammographic screening programs [9] and advances in treatment (e.g. improved radiotherapy techniques and new systemic therapeutic agents such as third-generation chemotherapy, aromatase inhibitors and HER2-targeted therapies). Clinical guidelines, as well as international position papers, state that surgery is cornerstone in early-stage BC patients [3,10]. Adjuvant chemotherapy is recommended for node-positive or high-risk node-negative disease, and endocrine therapy for high-risk receptor-positive disease [3,10]. However, population-based studies showed that older BC patients are less likely to receive standard care, which has been linked to lower survival rates in these patients [6,8,11].

Primary treatment for early-stage CRC patients is surgery. To reduce the risk of recurrence after surgery, clinical guidelines recommend that all lymph node-positive patients should be considered for adjuvant treatment, e.g. radiotherapy, or chemotherapy. Previous population-based studies showed that the administration of adjuvant chemotherapy has increased in recent years, and that this is associated with improved survival [12,13]. However, the administration of adjuvant chemotherapy is considerably less common in older patients [12].

Several studies state that age is not a contraindication for surgery or adjuvant chemotherapy in older patients [14]. Trends in relative survival and treatment have been published before, but this has not clarified the differences in survival rates between cancers. The aim of our population-based study was to investigate trends in survival rates and treatment patterns over time for older versus younger patients with BC and CRC in the Netherlands.

2.       The introduction should be revised to emphasize why CRC survival is being compared with BC survival, since the authors have cited previously published works in the introduction.

We revised the introduction of the manuscript, and emphasized why BC is compared with CRC.

Introduction, page 2

3.       Line 221:Since only female BC patients were included in the study, one would have expected that only female CRC patients will have been included too, in order to reduce any errors in result interpretation from confounders such as sex hormones. Survival can be greatly impacted by the sex of an individual. Table 1 shows that there were more males (53.5%) in the CRC group. How did you account for this in your study?

We agree with the reviewer that sex is an important confounder for survival, which was also demonstrated by the EUROCARE working group. We removed the male BC patients from the analyses (n = 768), as these numbers were much lower than the number of female BC patients. We adjusted Table 2 accordingly. Most importantly, the analyses were adjusted for sex and age. Hence, sex is not a confounder for the final results.

4.       The Ederer II Method and the Multivariate GLM were used to calculate 5 year RS and RER for individual age groups. An observed difference in survival of individual groups over time does not necessarily mean a true difference between group survival. A Cox regression model is required to determine if there are true differences in survival between the age groups. 

The study design is not compelling to analyse the overall survival. The aim of the study was to analyse trends of treatments and survival, which can be demonstrated with the Ederer II method and the Multivariate GLM. 

We agree with the reviewer that age is an important confounder for survival, and therefore stratified the results per age group. In addition, we adjusted for age and gender, to analyse the overall survival of BC and CRC patients.

Round 2

Reviewer 2 Report

The authors have responded to some of the concerns raised from the first review. However important aspects of the review were not responded to adequately.For example:

Line 221:Since only female BC patients were included in the study, one would have expected that only female CRC patients will have been included too, in order to reduce any errors in result interpretation from confounders such as sex hormones. Survival can be greatly impacted by the sex of an individual. Table 1 shows that there were more males (53.5%) in the CRC group. How did you account for this in your study?

The authors' response about excluding only male BC patients and including male CRC patients in the survival analyses has not addressed my major concern. In the first version of the manuscript, male BC patients were already excluded, therefore, nothing new was done in this regards. 

Also, there is a need for a Cox survival analysis in order to determine if a true difference in hazard ratios exists between the age categories as suggested by the aim of the study.

Round 3

Reviewer 2 Report

The authors have responded to my concerns from previous reviews, and the manuscript looks much improved. However, there are still a few corrections to be done.

Lines 54 and 72: Correct 'considerate' to 'considerably'.

Figure1: Give this a better title. Consider "Tumor stage by age at diagnosis". Also comment on Fig.1 in the results narrative, since it looks like an important aspect of the study.

Table 2: How do you interpret the HRs for years at diagnosis? Are you saying that those diagnosed for BC and CRC in 2012 are less likely to die compared to those diagnosed in 2003 (HR= 0.77 and 0.71 respectively)? I do not think that this makes much sense, and would suggest that you remove this from the model, and re-run the Cox regression.

Wishing you best of luck!
